

# Vocal communication in wild chimpanzees: a call rate study

Anne-Sophie Crunchant[1], Fiona A. Stewart[1,2] and Alex K. Piel[2]

[1] School of Biological and Environmental Sciences, Liverpool John Moores University, Liverpool, United Kingdom
[2] Department of Anthropology, University College London, London, United Kingdom

## ABSTRACT

**Background**. Patterns of vocal communication have implications for species conservation: a change in calling behaviour can, for instance, reflect a disturbed habitat. More importantly, call rate is a parameter that allows conservation planners to convert call density into animal density, when detecting calls with a passive acoustic monitoring system (PAM).

**Methods**. We investigated chimpanzee (*Pan troglodytes schweinfurthii*) call rate during the late dry season in the Issa Valley, western Tanzania by conducting focal follows. We examined the socio-ecological factors that influence call production rate of savanna woodland chimpanzees.

**Results**. We found that sex, proportion of time spent in a vegetation type, proportion of time spent travelling, time of the day, party size and swollen parous female presence had a significant effect on the call rate. Call rate differed among the different demographic classes with subadult and adult males vocalising twice as often as the subadult and adult females and three times as often as the juveniles.

**Applications**. The use of PAM and recent statistical developments to estimate animal density is promising but relies on our knowing individual call rate, often not available for many species. With the improvement in automatic call detection, we anticipate that PAM will increasingly be broadly applied to primates but also across taxa, for conservation.

## INTRODUCTION

Vocal communication is a means through which senders and receivers exchange information via the production of acoustic signals and is likely influenced by natural and sexual selection (*Seyfarth & Cheney, 2003*). Vocal signals are widely diversified among taxa, produced by insects, fish, herpetofauna, birds, and mammals to communicate in various social and environmental contexts, from alerting conspecifics to predator presence (*e.g., Schel et al., 2013*; *Vitousek et al., 2007*), maintaining bonds (*e.g., Fedurek et al., 2013*; *Wanker et al., 1998*), and marking territorial boundaries (*e.g., Peek, 1972*) among others. Similar to food, mates, and territory, acoustic space can also be a scarce resource for which animals compete. Callers must adjust spatial, temporal and frequency patterns in response to both abiotic and biotic factors, especially the sounds of sympatric fauna

Corresponding author
Alex K. Piel, a.piel@ucl.ac.uk

(*Araya-Salas et al., 2017*). Widely spaced individuals use long calls to maintain inter-individual contact, which allow them to coordinate movements, especially for socially fluid animals with high fission–fusion dynamics like elephants (*Loxodonta africana*) (*e.g.*, *Leighty et al., 2008*), spotted hyena (*Crocuta crocuta*), (*e.g.*, *Theis et al., 2007*), bottlenose dolphins (*Tursiops truncatus*) (*Janik & Slater, 1998*), beluga whales (*Delphinapterus leucas*) (*O'Corry-Crowe et al., 2020*), spider monkeys (Ateles spp.) (*e.g.*, *Spehar & Di Fiore, 2013*, bonobos (*Pan paniscus*) (*e.g.*, *Hohmann & Fruth, 1994*) and chimpanzees (*e.g.*, *Fedurek, Donnellan & Slocombe, 2014*).

Call rate, the number of calls emitted per unit time per individual, can advertise male quality (*Pedroso et al., 2013*)  and rate changes with caller age and sex class, time of day, group size and composition, social context, and environment, among others (*Pérez-Granados et al., 2019*). For instance, frog vocalizations are highly sexually different, energetically costly, and are mostly produced by males (*Emerson & Boyd, 1999*). Dawn and dusk chorusing are common for many species, exploiting low ambient and minimal wind noise levels (*Ey & Fischer, 2009*). Guereza black and white colobus monkey (*Colobus guereza*) calls are highly contagious and spread from one group to another (*Schel & Zuberbühler, 2012*). Furthermore, loud calls in baboons advertise male quality, with high-ranking males calling more often (*Kitchen et al., 2003*; *Fischer et al., 2004*). Therefore, loud calls serve multiple functions and spatiotemporally shift in predictable ways, which can be reflected in a variation of call rates.

Additionally, vocal communication has implications for species conservation. A change in calling behaviour can reflect a disturbed habitat. Anthropogenic pressure affects call parameters, such as call duration and call frequency, and also the number of calls produced. For instance, a recent study at Los Tuxtlas in Mexico showed that howler monkeys (*Alouatta palliata mexicana*) produced fewer calls when exposed to anthropogenic noise (*Cañadas Santiago et al., 2020*). There is also evidence that African elephants change the acoustic structure of their alarm calls when threatened by bees or humans (*Soltis et al., 2014*). *LaZerte, Otter & Slabbekoorn (2017)* demonstrated that male mountain chickadees (*Poecile gambeli*) adjust songs, calls, and chorus composition with increasing ambient and (experimental) anthropogenic noise. More importantly, call rate is a parameter that allows conservation planners to convert call density (the number of calls per unit time per unit space) into animal density (*i.e.*, the number of callers per unit space). Acoustic spatial capture-recapture (aSCR) and distance sampling methods can be used to estimate animal density by detecting vocalisations with acoustic sensors (*e.g.*, *Marques et al., 2013*; *Stevenson et al., 2015*). This can be particularly useful for cryptic, elusive, wide ranging and not visually detectable species. Using this method, numerous studies have reported abundance and density for various species, such as ovenbirds—*Seiurus aurocapilla* (*Dawson & Efford, 2009*; *Efford, Dawson & Borchers, 2009*), frogs—*Arthroleptella lighfooti* (*Borchers et al., 2015*; *Stevenson et al., 2015*), northern yellow-cheeked gibbons—*Nomascus annamensis* (*Kidney et al., 2016*). We can convert caller density to population density once we know the call rate (*Stevenson et al., 2015*). Data on these parameters can be obtained by following individuals and recording call events. Obtaining these values is what motivated the current study. To accurately reflect the call rate of calls that can be detected on acoustic
sensors, sometimes deployed hundreds of meters away from the caller for subsequent analyses, we focused only on screams, barks and pant hoots.

Chimpanzees (*P. troglodytes*) are a gregarious species and form small parties that change in size and composition throughout the day. They move through a relatively large territory—*e.g.*, from 7 to 40 km$^2$ for forest dwelling chimpanzees (*Newton-Fisher, 2003*; *Després-Einspenner et al., 2017*) and 72 to 90 km$^2$ for savanna dwelling chimpanzees (*Samson & Hunt, 2012*; *Pruetz & Herzog, 2017*) and are often hundreds of metres apart. Thus, they must rely on vocal communication to reveal to others information about, for instance, fruiting trees, predator presence, and movement coordination within and between parties. *Eckhardt, Polansky & Boesch (2015)* reported that male chimpanzees from Taï Forest (Ivory Coast) remained out of visual range of conspecifics for almost half of observation time but within auditory range (<1 km) for 70% of the time, suggesting chimpanzee vocalisations serve an important function as contact calls between spatially separated individuals. Some of the earliest studies on chimpanzee vocalisations noted an overall sex difference in pant hooting rates in adults, with males calling more than females (*Marler & Hobbett, 1975*; *Goodall, 1986*), and in some cases, all-female parties calling so rarely to make inter-sex comparisons impossible (*Clark, 1996*; *Clark & Wrangham, 1994*). Only recently, however, were these observations supported with empirical data from a single community in Tai Forest (*Kalan, 2019*). Thus, despite a half-century of investigation into chimpanzee vocalisations, much is still not known about call rate, and especially how it changes between sexes, behavioural contexts, and communities. This intra- and inter-community rate variability has bearing on what value is used for density studies that rely on PAM data, where call rate is a critical parameter (*Marques et al., 2013*).

The aim of this study was twofold: first, we wanted to estimate the call rate of chimpanzees living in a savanna woodland habitat. Given the known sex differences in chimpanzee acoustic communication, *e.g.*, call rate and acoustic parameters (*Kalan, 2019*), we calculated the call rate for the following age/sex classes: subadult and adult females, subadult and adult males and male and female juveniles. Second, we investigated socio-ecological factors influencing call rate. We therefore examined the effect of (1) party size: we expected chimpanzees to call more often as party size increased due to the chorus effect and contagious calling (*Fedurek, Schel & Slocombe, 2013*); (2) presence of a swollen female: we expected that call rate would be higher when callers were in parties with at least one parous swollen female (*Fedurek, Donnellan & Slocombe, 2014*); (3) time of day: we expected a temporal pattern with chimpanzees calling more often in the morning and late afternoon (*e.g.*, *Piel, 2018*); (4) vegetation: chimpanzees seem to spread more when in open area, likely because of higher visibility and we consequently expected chimpanzees to call more often when present in the open vegetation (woodland); (5) activity: two important call functions are to indicate fruiting trees and maintain spatial cohesion, especially during travelling, hence we expected that the proportion of time spent travelling or feeding would be positively correlated to the call rate (*Clark & Wrangham, 1993*; *Fedurek, Donnellan & Slocombe, 2014*).

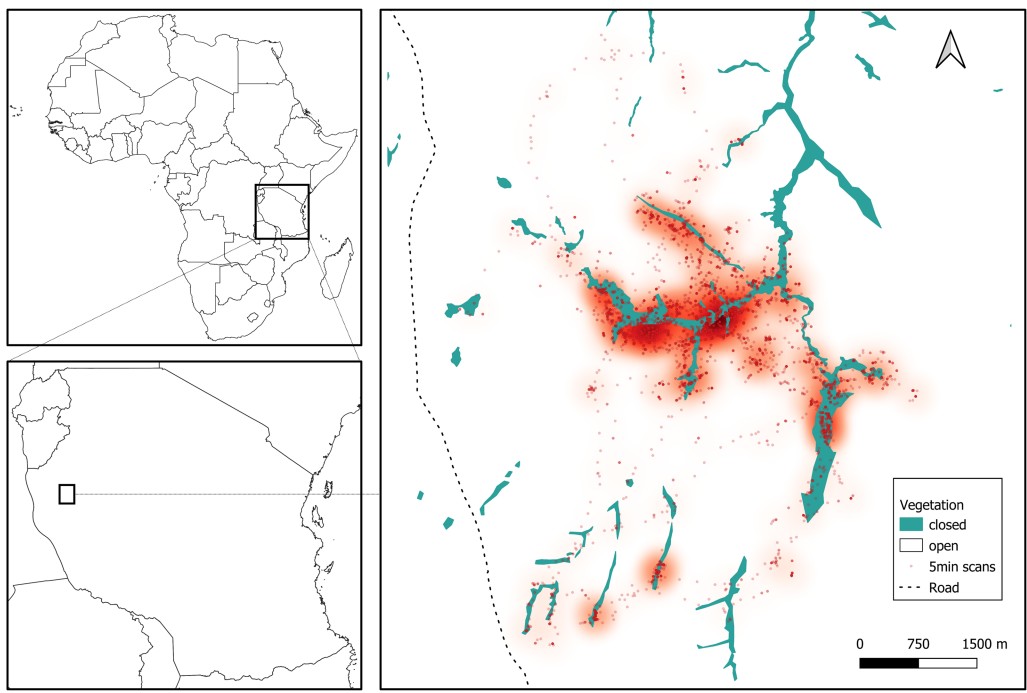

**Figure 1** **Study site in Issa Valley, Western Tanzania and chimpanzee locations during the late dry season from focal follows (7239 5-min scans).** Each dot represents the location of an individual and a heatmap shows the preferentially used areas during the study period.

## MATERIALS & METHODS

### Study site

We collected data for three months during the late dry season, between August and October 2019, in the Issa Valley, western Tanzania (Fig. 1). The study area of about 70 km$^2$ is comprised of a series of valleys separated by steep mountains and flat plateaus, with an altitudinal gradient ranging from 1,050 to 1,650 m above sea level. Vegetation is dominated by miombo woodland and also includes grassland, swamp and riparian forest. For analyses, we collapsed vegetation categories into 'open' (woodland, grassland, swamp) and 'closed' (riparian forest), see Fig. 1. The region is one of the driest and most open that is inhabited by chimpanzees (*Moore, 1992*) and characterised by two seasons: wet (November to April) and dry (May to October). Annual rainfall averaged 1220 mm per annum (range from 930 to 1,490 mm from 2009 to 2014) and temperatures ranged from 11 °C to 38 °C (*Piel et al., 2017*).

### Study subjects

The study site covers the territory of at least two chimpanzee communities (one habituated community and one or more neighbouring communities). When the study began, the habituated Issa community comprised nine adult females, seven adult males, four subadult males, one subadult female, three juveniles and four infants (Table 1). During the study period, one female (AZ) was not yet fully habituated so we have not included data on her,

**Table 1 Summary of the number of focal follow hours per individual and their call rate.** Mean call rate, defined as the number of pant-hoots, screams or barks per hour is presented with the individual range in brackets.

| Name | Age | Sex | # h of follow | Call rate (# calls/hr, min–max) |
|------|-----|-----|---------------|---------------------------------|
| AZ | Adult | Female | 0 | NA |
| BS | Adult | Female | 13 | 0.38 [0–2] |
| JU | Adult | Female | 11 | 1.73 [0–4] |
| KL | Adult | Female | 5 | 0.60 [0–2] |
| KN | Adult | Female | 9 | 2.22 [0–5] |
| KJ | Adult | Female | 14 | 1.14 [0–3] |
| MA | Adult | Female | 18 | 1.56 [0–5] |
| ZA | Adult | Female | 17 | 0.24 [0–2] |
| KS | Sub-adult | Female | 26 | 0.31 [0–3] |
| BG | Adult | Male | 41 | 3.41 [0–11] |
| EL | Adult | Male | 23 | 2.04 [0–8] |
| IM | Adult | Male | 35 | 2.37 [0–9] |
| KT | Adult | Male | 37 | 2.65 [0–14] |
| MY | Adult | Male | 33 | 1.06 [0–6] |
| SM | Adult | Male | 35 | 2.23 [0–9] |
| WA | Adult | Male | 23 | 1.48 [0–8] |
| DH | Sub-adult | Male | 25 | 2.60 [0–7] |
| MS | Sub-adult | Male | 29 | 1.45 [0–6] |
| SN | Sub-adult | Male | 30 | 2.90 [0–13] |
| WG | Sub-adult | Male | 30 | 1.53 [0–8] |
| MW | Juvenile | Female | 5 | 1.20 [0–2] |
| KK | Juvenile | Male | 28 | 0.57 [0–5] |
| BN | Juvenile | Male | 0 | NA |

as researcher presence likely influenced her natural behaviour (*Crofoot et al., 2010*; *Nowak et al., 2014*). Two individuals (adult female and infant) were killed and a female gave birth during the study period. The habituated community has a home range $\geq 55$ km$^2$.

## Data collection

We selected a focal chimpanzee (adult, subadult or juvenile) each morning and tried to follow him/her for the entire day *i.e.*, from nest to nest). We conducted instantaneous focal animal sampling, with a scan defined as the behaviour of the animal recorded every five minutes, when we collected data on caller location (GPS), behaviour (travelling, feeding, resting, grooming, playing, other), vegetation (open or closed) and party size, defined as the number of juveniles, subadults and adults seen. We further noted all vocal behaviour ad libitum of the focal, including the type of vocalisation (pant grunt, grunt, hoot, pant hoot, bark, scream or combinations of different types—see *e.g.*, *Crockford, 2019*; *Goodall, 1986* for descriptions of chimpanzee vocalisations). Successive calls were considered as new events when separated by more than one second. We included only vocalisations that involved at least a scream, a pant-hoot or bark in the analyses, to match the calls that can be

potentially recorded by an acoustic recorder deployed about 500 m away from the caller. We thus excluded grunts and other closed calls that do not propagate far. We included only hours of follow data with at least 10 scans/hour, which corresponds to at least fifty minutes of observation per hour.

### Data analyses

We conducted all analyses in R v.3.6.1 (*R Core Team, 2019*). For each hour of focal follow, we determined the proportion of time spent travelling, feeding and other behaviours, in open or closed vegetation by the focal individual. We defined the proportion of time in *e.g.*, open vegetation as the number of scans in which the focal was observed in open vegetation divided by the total number of scans where the focal was in view.

To model the number of calls per hour (call rate, CR) as a function of the covariates, we used a negative binomial distribution GAM with a log link function, that allowed us to account for overdispersion. Fixed covariates were (1) time (T, categorical with 11 levels: from 7am to 6pm), (2) mean hourly party size (PS, continuous), (3) presence of swelling female in the party (PrS, categorical with 2 levels: presence or absence), (4) proportion of time spent in closed *vs.* open vegetation (F, continuous), (5) proportion of time spent feeding (Fe, continuous), (6) proportion of time spent travelling (Tr, continuous) and (7) age-sex class (AS, categorical with 3 levels: adult-subadult female, adult-subadult male, juvenile). To incorporate the dependency among follows on the same day, we used 'individual' as a random intercept and to incorporate the dependency among observations of the same individual at the same time, we used "date" as a random intercept. We used the package mgcv (*Wood, 2017*) to fit the model. We centred and scaled continuous predictors.

We tested predictors for collinearity by calculating variation inflation factors (VIF) using the package car (*Fox & Weisberg, 2018*) in an equivalent linear model including only the fixed effects from each model fitted with the package MASS (*Brian et al., 2020*). Multicollinearity was not present (maximum VIF: PS = 1.53). We verified model assumptions by plotting residuals versus fitted values and QQ-plots. We ran a set of models and ranked them by AICc value.

## RESULTS

In total, we analysed 487 h of follows on twenty-one chimpanzees (21.2 ± 12.2 h per individual, Table 1). Call rate varied individually, ranging from an average of 0.24 to 3.41 calls per hour.

### Factors influencing the call rate

We did model averaging among models with ΔAICc < 2. The significant effects in the best averaged model are age/sex, proportion of time in the forest, proportion of time travelling, time-of-day, presence of swollen female, party size, and the two random intercept terms (individual and date) (Table 2).

Results of the GAM revealed that the age-sex class had a significant effect on call rate: overall, Issa chimpanzees exhibited a mean call rate of 1.91 with a 95% CI of [1.52−2.40]) calls per hour for the subadult and adult males, 0.84 with a 95% CI of [0.59−1.21] calls

**Table 2 Model selection.**

| Model | df | logLiK | AICc | delta | Weight |
|---|---|---|---|---|---|
| $CR \sim AS + PrS + F + Tr + PS + T + F$ | 24 | −814.710 | 1681.5 | 0.00 | 0.094 |
| $CR \sim AS + PrS + F + Tr + PS + T + F + D$ | 24 | −814.710 | 1681.5 | 0.00 | 0.094 |
| $CR \sim AS + PrS + F + Tr + T + F$ | 25 | −814.471 | 1682.4 | 0.84 | 0.062 |
| $CR \sim AS + PrS + F + Tr + T + F + D$ | 25 | −814.471 | 1682.4 | 0.84 | 0.062 |
| $CR \sim AS + PrS + Fe + F + Tr + PS + T + F$ | 25 | −814.078 | 1682.6 | 1.09 | 0.055 |
| $CR \sim AS + PrS + Fe + F + Tr + PS + T + F + D$ | 25 | −814.078 | 1682.6 | 1.09 | 0.055 |
| $CR \sim AS + F + Tr + T + F$ | 24 | −816.013 | 1683.0 | 1.50 | 0.045 |
| $CR \sim AS + F + Tr + T + F + D$ | 24 | −816.013 | 1683.0 | 1.50 | 0.045 |
| $CR \sim PrS + F + Tr + T + F$ | 26 | −813.445 | 1683.4 | 1.88 | 0.037 |
| $CR \sim PrS + F + Tr + T + F + D$ | 26 | −813.445 | 1683.4 | 1.88 | 0.037 |
| $CR \sim PrS + Fe + F + Tr + T + F$ | 26 | −812.771 | 1684.1 | 2.54 | 0.027 |
| $CR \sim PrS + Fe + F + Tr + T + F + D$ | 27 | −812.771 | 1684.1 | 2.54 | 0.027 |

**Notes.**

CR, call rate; AS, age/sex class; PrS, presence swelling female; Fe, proportion of time spent feeding; F, proportion of time spent in closed vegetation; Tr, proportion of time spent travelling; PS, party size; T, time; F, follow; D, date.

**Table 3 Outcome of a GAM investigating the effect of time, age/sex class, presence of swollen parous female, party size (PS), proportion of time spent in the closed area and proportion of time spent travelling on chimpanzee call rate for the averaged best models.**

| Predictors | Parameter estimate | | | |
|---|---|---|---|---|
| | Estimate | Std. E. | z value | Pr(>\|z\|) |
| Intercept | 0.477 | 0.225 | 2.115 | 3.44e−02[*] |
| Age/sex class (male) | | | | |
| *Subadult/adult female* | −0.864 | 0.252 | 3.413 | 6.43e−04[**] |
| *juvenile* | −1.129 | 0.456 | 2.820 | 4.79e−03[**] |
| Closed area (forest) | −0.171 | 0.063 | 2.698 | 6.977e−03[**] |
| Swollen parous female presence | 0.315 | 0.141 | 2.236 | 2.536e−01[*] |
| Party size | 0.164 | 0.734 | 2.223 | 2.619e−02[**] |
| Travel | 0.237 | 0.059 | 3.977 | 1.60e−04[***] |

**Notes.**

Parameter estimates are reported for all terms in the averaged best three models.

[*]$p < 0.05$.
[**]$p < 0.01$.
[***]$p < 0.001$.

per hour for the subadult and adult females, and 0.50 with a 95% CI of [0.24−1.05] calls per hour for the juveniles. With 95% confidence, males call between 1.5 and 3.8 times as frequently as females and between 1.3 and 7.9 times as frequently as juveniles. Chimpanzees vocalised significantly more as the proportion of time spent in open vegetation and as the proportion of time spent traveling increased (Table 3). The smooth effect of time reveals call rates being highest in the morning decreasing thereafter, before increasing late afternoon (Fig. 2).

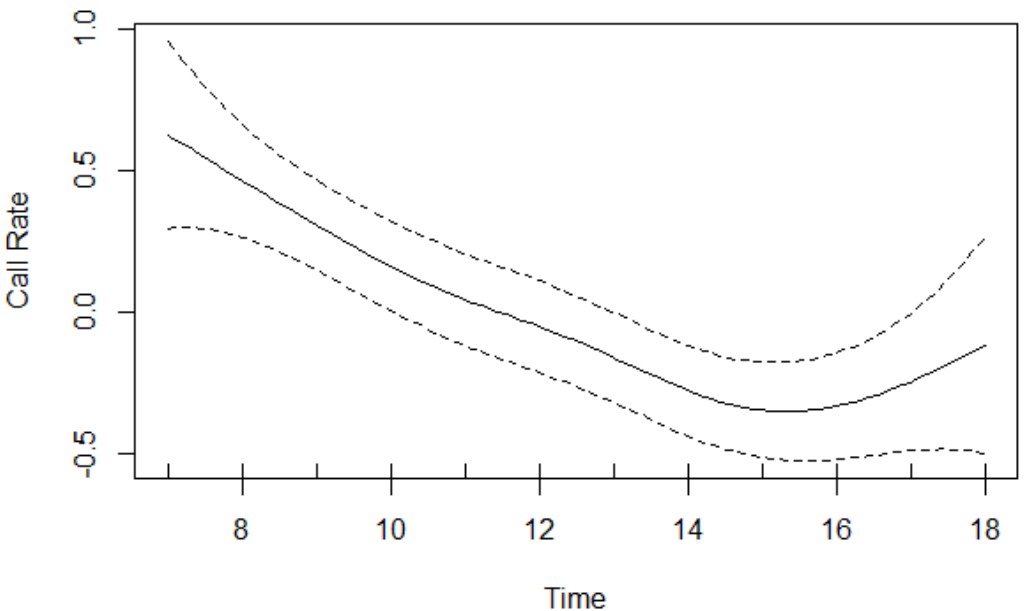

**Figure 2** **Call rate changes over the course of the day.** Rates were highest in the morning and decreased thereafter, before increasing in the late afternoon.

## DISCUSSION

In this study, we sought to establish the mean call rate of different demographic classes of wild chimpanzees, as well as examine the socio-ecological factors that influence call production rate. We found that time spent in a specific vegetation type, time spent travelling, time of the day, party size and swollen parous female presence had significant effects on call rate.

### Call rate among demographic classes

Our results confirm early reports on sex differences between adult chimpanzee loud call production (*Clark, 1996*) and a recent study that subadult and adult male chimpanzees more than twice as often as subadult and adult females (*Kalan, 2019*). The sexual dimorphism seems to be even more pronounced for forest than savanna woodland dwelling chimpanzees. At Taï, males produced on average $2.5 \pm 1.08$ calls per hour and females produced $0.88 \pm 0.32$ calls per hour (*Kalan, 2019*), compared to $1.91 \pm 0.12$ calls per hour for the males and $0.84 \pm 0.18$ calls per hour for the females at Issa (this study). We propose two explanations for this sex difference: (1) necessity of maintaining strong bonds and/or (2) sexual selection. Chimpanzees are male-philopatric, where males stay in their natal territory, while females disperse at sexual maturity and migrate to neighbouring communities (*e.g., Moore, Langergraber & Vigilant, 2015*; *Nishida, 1968*). This sex difference in dispersal explains strong male-male relationships (*e.g., Mitani, 2009*). Males tend to develop stronger bonds with other male community members with whom they spend more time, forming a linear dominance hierarchy and developing affiliations and coalitions. East African female chimpanzees, on the contrary, are far less gregarious

than males, travel less, and spend most of their time alone or with their offspring in their core home range (*Wrangham & Smuts, 1980*).

## Ecological factors

We found that chimpanzees called most at two times of day (morning and evening) and from open (miombo woodland) compared to closed (riparian forest) vegetation. This temporal information is important for conservationists, who can for instance record sounds with PAM systems at these periods to maximise the likelihood of detecting calls and simultaneously limit the number of audio files to analyse. There are ecological and social reasons for these patterns. Temporally, the bimodal pattern that we found in this study is similar to what has been reported previously at Issa (*Piel, 2018*) and elsewhere (*e.g., Wilson, Hauser & Wrangham, 2007*; *Wrangham, 1975*). Vocalisations allow parties to coordinate movements, notably prior to arrival at their nesting sites (*Fischer & Zinner, 2011*) or feeding trees (*Clark & Wrangham, 1993*). Ecologically, callers may exploit ideal sound transmission conditions. The Acoustic Adaptation Hypothesis predicts that animals may adjust their vocal signals to maximize signal transmission and minimize sound degradation, specifically within each environment in which calls occur (*Waser & Waser, 1977*; *Ey & Fischer, 2009*; *Brown & Waser, 2017*). Environmental metrics, such as temperature gradients, humidity and wind, vary with time of day, and can degrade signals, resulting in structural changes to the primary sound (*Waser & Brown, 1984*). Relatedly, the natural habitat distorts signals as distance from the sound origin increases. Open vegetation, with increased wind-induced noises compared to closed vegetation, can further degrade signals but sounds attenuate faster in closed vegetation because of tree density (*Brown & Waser, 2017*). If chimpanzees rely more on close contact calls (not included in this study) to maintain party cohesion in closed, rather than open vegetation, the fact that chimpanzees vocalize significantly more often in open vegetation might be explained by their activity, rather than the external environment. Furthermore, the Issa landscape is dominated (>65%) by miombo woodland vegetation, thus it is unsurprising that they vocalize more from open than closed vegetation types. Calls made in open vegetation could serve multiple purposes. First, individuals may spread out more in these areas, given that food sources are more widely distributed here than in closed areas and so calls are functioning to preserve party cohesion. Additional data on inter-individual distance (*e.g.*, party spread) in each vegetation type would help us resolve this. Second, calls travel further in open compared to closed vegetation, with fewer trees to attenuate sound (although see Waser & Brown, 2017), so individuals also may be calling to communicate with distantly located parties. Often counter calls are heard from these woodland pant hoots (A. Crunchant, 2019, pers. obs.), but we do not have comparative data from forests.

## Travelling and feeding activities

We evaluated the effect of activity, namely time spent travelling and feeding, on call production rate. While previous studies have shown that chimpanzees frequently pant-hoot at food sources (*Clark & Wrangham, 1993*; *Fedurek, Donnellan & Slocombe, 2014*; *Wrangham, 1977*), call rate did not change with the proportion of time spent feeding in

our study. However, our brief field season took place during the late dry season, when fruit availability is highest (*McLester et al., 2019*). The abundance of food might reduce competition among individuals and thus depress the need for calling. Furthermore, we did not incorporate which food items were consumed. Fedurek and colleagues (*2014*) have shown that male chimpanzees were more likely to pant hoot at high-quality food patches and thus we suggest that subsequent studies consider food type consumption during calling bouts.

The number of calls increased as the proportion of time spent travelling increased. It has long been demonstrated that loud calls facilitate fusion events and help regulate grouping dynamics and coordination among community members during travel (*e.g.*, *Goodall, 1986*). In other communities, males travel significantly more after a pant hoot is produced (*Mitani & Nishida, 1993*) and males are more likely to repeat a call prior to, rather than after, fusion with other males (*Fedurek, Donnellan & Slocombe, 2014*).

## Party size and presence of parous swollen female

We have shown that call rate was positively related to party size. Coordinating movement with a higher number of individuals requires more communication, especially for decision making and coordinating party fusion (*Fischer & Zinner, 2011*). Furthermore, a previous study has shown that the number of aggressive events is positively correlated to the number of males in a party (*Muller, 2002*), and thus the number of calls produced during agonistic events (*i.e.*, barks, screams and pant hoots) would likely increase with the number of males in the party.

Finally, the presence of a swollen female in the party did also impact call production rate, as expected. It has indeed been shown that males prefer to mate with parous females and are more involved in male-male competition in their presence because parous females are attractive to males, which exhibit aggressive displays and courtship behaviour during these times (*Muller, Thompson & Wrangham, 2006*).

## Implications and limitations

Variability in chimpanzee calling is well known, described in nearly every study on the topic; early work attributed variability to age, sex, context, party size, and community, among others (*Goodall, 1986*; Arcadi & Wrangham, 1993; Arcadi & Wrangham, 1994; Mitani et al., 1992; Mitani et al., 1994). For the most part, these studies reported variability in the context of call function. In the current study, however, we were not interested in call function. Whilst similar to some previous work, we report on context-specific variability, our primary interest is in the implications of call rate variability for conservation studies that rely on PAM to extract animal densities. For example, sex ratio, vegetation types and proportions, and activity budgets vary between communities. Our results here begin to reveal how call rate changes with each of these, which has direct bearing on how its parameterised in density analyses. Specifically, for communities that range across more open vegetation - Issa (Tanzania), Fongoli (Senegal), Semliki (Uganda) - call rate should reflect this in density models, varying with sensor location.

Those implications notwithstanding, we are cautious interpreting these results and conclusions should not be generalized to scenarios other than that where the data were
collected. We sampled over a single and brief 3-month season and recorded only 487 h of follows (compared to *Kalan (2019)* who recorded 731.5 h of follows across 10 months). With food availability, party size and the number of swollen females highest at this time of year (A. Piel, 2021, unpublished data), it is likely that our data are inflated against annual means. More data collected across multiple seasons would resolve this uncertainty and would result in seasonal variation in call production, especially lower call rates in the early wet season when party size declines and fewer females exhibit maximum tumescence. We were also unable to equally balance data collection across the community, with some individuals being followed only 5 h and some more than 30 h. This was mostly due to the difficulty of following recently habituated females. Given that the call rate variation between two individuals is likely to be greater than the variation within the same individual, rather than over-sample fewer individuals, we chose to sample more individuals.

## Future directions

Future studies can not only build off these results by adding more data, but also additional predictor variables that may influence chimpanzee call rate. Specifically, further analyses would benefit from evaluating the effect of rank on call rate. For instance, rank has been implicated in influencing call rate in chacma baboons (*Papio cynocephalus ursinus*) (*Kitchen et al., 2003*), gelada monkeys (*Theropithecus gelada*) (*Benitez et al., 2016*), orang-utans (*Pongo pygmaeus*) (*Mitani, 1985*) and also non-primate (fallow bucks (*Dama dama*) (*Pitcher et al., 2014*)) species, and strongly suggests that vocal communication is influenced by sexual selection. In support of this, multiple studies examining chimpanzee rank and call production reveal a positive relationship between male quality, testosterone, and rank (*Clark, 1993*; *Clark & Wrangham, 1993*; *Mitani & Nishida, 1993*; *Fedurek et al., 2016*). There is not yet empirical evidence however, demonstrating female preference for male vocalisations.

Moreover, we recommend that subsequent studies include a spatial component. For example, chimpanzees spend the majority of their time in the core of their home range, usually representing about 75–90% of their total territory (*Wilson, Hauser & Wrangham, 2007*). When males conduct patrols in high risk areas, *e.g.*, croplands or territorial boundaries, call rate declines significantly (*Wilson, Hauser & Wrangham, 2007*). Chimpanzees from savanna habitats have far larger home ranges—*e.g.*, $\geq 55$ km$^2$ at Issa (C. Giuliano, 2021, unpublished data) or about 90 km$^2$ at Fongoli (*Pruetz & Herzog, 2017*)—than forest dwelling chimpanzees—*e.g.*, 7 km$^2$ at Budongo (*Newton-Fisher, 2003*). *Suzuki (1969)* proposed that the low density and widely distributed foods in these savanna landscapes may promote more nomadism in savanna than in forest chimpanzees and *Moore (1992)* suggested that the sheer scale of these home ranges may make them indefensible. Forest-dwelling chimpanzees are well known to be highly xenophobic and respond aggressively to members from neighbouring communities (*e.g.*, *Goodall, 1986*; *Mitani, Watts & Amsler, 2010*; *Mitani & Watts, 2005*; *Watts & Mitani, 2001*; *Wilson & Wrangham, 2003*). In contrast to forest dwelling chimpanzees, recent observations at Issa suggest that chimpanzees there may be more tolerant of neighbouring individuals, sharing large parts of their territory with non-community members (pers. obs). Similarly, as

mentioned in the introduction, many species modulate their call rate in response to human disturbance—*e.g.*, amphibians (*Sun & Narins, 2005*), birds (*LaZerte, Otter & Slabbekoorn, 2017*), and elephants (*Soltis et al., 2014*). For example, in Bili, northern DRC, chimpanzees call significantly less when they nest near human settlements (*Hicks & Roessingh, 2010*). The Issa study site lies not far from a human settlement and a road. Despite the fact that snare encounter rates increased with distance from the research station (*Piel et al., 2015*), the presence of cattle herding is still frequently observed in the area (AC, pers. obs.). We recommend that subsequent work evaluates whether Issa chimpanzees adjust their call rate near areas of increased human presence (*e.g.*, road or cattle herders).

## CONCLUSIONS

Whilst we discussed the inherent importance of biological and social predictors of call rate in wild chimpanzees, this study was primarily motivated by the need to establish call rate to estimate chimpanzee density from passive acoustic monitoring (PAM) and acoustic spatial capture-recapture (aSCR) methods (A. Crunchant, 2021, unpublished data). Chimpanzee call rate is highly sexually dimorphic and like many other chimpanzee behaviours (*e.g.*, *Kühl et al., 2019*; *Whiten et al., 1999*) shows community-specific patterns. To estimate an average and unbiased call rate, and consequently an unbiased density estimate, we need to weight values of each age/sex class by the proportion of each demographic class constituting the community. Similar to great ape nest decay and nest production rates, even context-specific call rate is likely to vary between communities. We recommend call rate studies to be conducted in parallel to PAM deployment. Call rates can only be studied in habituated chimpanzees, however, as follows are necessary to add context to vocalizing behaviour. Comparison of call rates between communities will be instrumental to evaluate how strongly it can affect density estimation from aSCR methods. The use of PAM and aSCR to estimate chimpanzee density is promising and with the improvement of automatic call detection, we anticipate that PAM will become more common in the primatologist's toolbox.

## ACKNOWLEDGEMENTS

We are also extremely grateful to all field assistants of the Greater Mahale Ecosystem Research and Conservation (GMERC) project and Caroline Fryns for their help in the field. Many thanks to Edward McLester for statistical advice and Ineke Knot, Ben Stevenson, Barbara Fruth, Tiago Marques and one anonymous reviewer for feedback on a previous version of the manuscript.

### Funding

Data collection was supported by the International Primatological Society through the Conservation grant, and Liverpool John Moores University. Long term funding for ongoing research at Issa is supported by the UCSD/Salk Center for Academic Research

and Training in Anthropogeny (CARTA). The funders had no role in study design, data collection and analysis, decision to publish, or preparation of the manuscript.

## Grant Disclosures

The following grant information was disclosed by the authors:

International Primatological Society.

UCSD/Salk Center for Academic Research and Training in Anthropogeny (CARTA).

## Competing Interests

The authors declare there are no competing interests.

## Author Contributions

- Anne-Sophie Crunchant conceived and designed the experiments, performed the experiments, analyzed the data, prepared figures and/or tables, authored or reviewed drafts of the paper, and approved the final draft.
- Fiona A. Stewart conceived and designed the experiments, authored or reviewed drafts of the paper, instrumental in habituation of chimpanzees and maintenance of research station that hosted the study, and approved the final draft.
- Alex K. Piel conceived and designed the experiments, performed the experiments, authored or reviewed drafts of the paper, and approved the final draft.

## Animal Ethics

The following information was supplied relating to ethical approvals (i.e., approving body and any reference numbers):

The Tanzanian Wildlife Research Institute (TAWIRI) and Commission for Science and Technology (COSTECH) approved this work in Tanzania, as did Liverpool John Moores University.

## Field Study Permissions

The following information was supplied relating to field study approvals (i.e., approving body and any reference numbers):

Field experiments were approved by GMERC, which oversees research in the sea valley, Tanzania.

## Data Availability

The data are available in a Supplementary File and at figshare: Chitayat, Adrienne; Piel, Alexander; Stewart, Fiona; Lewis, Matthew J.; Wich, Serge A. (2020): Ecological correlates of chimpanzee (Pan troglodytes schweinfurthii) density in Mahale Mountains National Park, Tanzania. figshare. Collection. https://doi.org/10.6084/m9.figshare.c.5136293.

## Supplemental Information

Supplemental information for this article can be found online at http://dx.doi.org/10.7717/peerj.12326#supplemental-information.

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
