# Peer review of "Vocal communication in wild chimpanzees: a call rate study"

_PeerJ, doi:10.7717/peerj.12326_

## Round 0.1 · original submission · Minor Revisions

Both reviewers were positive, and both had many suggestions for changes in your report. Please revise the Ms according to both reviewers (I did not see disagreements between the two...). The last comments by reviewer #1 ask for a discussion of overall significance, especially the value of your work to conservation. In my experience, this is often the hardest thing to do in a paper, but it's well worth the effort, as your paper will have greater impact when the significance (e.g., value to conservation) is clearly and prominently presented.

Reviewer 1 ·

Basic reporting

The manuscript is concise and is well written for the most part, and the authors make their arguments clearly. I have just a few minor criticisms of the presentation:

In the Introduction, the authors state that PAM allows conversion of “call density” into “animal density”. What is “call density”? Density is a spatial measure, and readers can infer that “animal density” means population density (in this case, the population density of chimpanzees in the Issa region), but I don’t think it make sense to talk about the “density” of pant hoots.

46: Chimpanzees don’t “live in a fission-fusion structure”; they have high fission-fusion dynamics.

52: What does “inform on fitness” mean? And precisely how could pant hooting rate be associated with fitness, or be a reliably indicator of the caller’s quality as a mate?

96-97: “in function of sex” – this would be better as something like “A sex difference in pant hooting rates exists in adults, with males calling more than females, and high-ranking males call more often than low-ranking males”.

103: “Sexual dimorphism” -- just “sex differences” is better, but if you want to use a technical term, the appropriate one is “sexual diethism”.

113: “A higher visibility”: omit “a”

115: You say “one important function”, but then give two functions. Re-phrase this.

138: “was comprised of nine…” should be “comprised nine…”

231: This should be “sex differences in dispersal”, not “sexual dispersal”

239: “Male-male competition to attract females is primordial” – “primordial” is a strange term in this context. “Primary”? “Most important”? However, I disagree that male competition in chimpanzees is mostly about “attracting” females. This implies that it functions mostly to influence female mate choice, but female choice seems to have limited importance in chimpanzees other than its influence on whether male consortship attempts succeed (although the small size of the study community and small number of adult males may provide females there with more potential to exercise choice than is typical).

314: “Given that inter-individual call rate is likely to be greater than within-individual rate” -- ? Something is missing here. Do you mean that “variation” in call rate is probably greater among than within individuals? If this is what you want to say, what do you mean by “within-individual variation” in call rates?

Experimental design

The authors have explained their methods and I have no serious concerns about the statistical methods. However, I have one question about definitions and another more substantive one about the methods.

First: Did you categorize “arrival” pant hoots (those given on arrival at a food source) as associated with “traveling” or with “feeding”? How did you determine when a pant hoot was associated with feeding?

Initially, I wondered why you talked about the correlations between calling rates, on the one hand, and the “proportions” of time spent feeding or traveling, on the other, and did not simply ask whether rates were higher while the chimpanzees were feeding and/or traveling than they were when the chimpanzees were engaged in other activities. Then I realized that you used each hour of a focal follow as a data point, and that in these analyses, the dependent variable was the number of calls per hour and the predictor variables were the % of the hour the focal subject spent feeding or spent traveling. This does not measure calling rates during feeding or traveling; calculating those measure would require counting the number of calls the individual gave while he or she was actually feeding and dividing by the time they were feeding, and doing the same for calls they gave while they were actually traveling. It is feasible, for example – if not terribly likely – that a focal individual calls 10 times in an hour during which they feed for 30 minutes, but none of the calls occur while they are actually feeding. If you re-did the analysis to examine call rates per hour spent feeding or traveling (instead of instead of rates per hour in relation to the 5 of the hour they were feeding or traveling), the results might change little, but this seems to me to be the proper way to address the question.

Validity of the findings

I don’t have any concerns regarding the validity of the findings per se other than my question above about rates of calling while individuals were feeding or traveling.

A couple of comments about the Discussion, though:

238: Delgado did not actually show that male orangutan long calls “advertise fitness”. Fitness is a measure of relative reproductive success (plus any indirect benefits derived via kin selection). Data on long calling are consistent with the hypothesis that the calls provide information about male quality, and the very limited paternity data show that flanged males, who give the calls, generally have higher reproductive success than unflanged males, but this does not mean that calls advertise fitness.

245: Why call pant hoots “agonistic calls”?

Also (see below), I think the authors need to more to show that their findings are novel – or perhaps better, to show why their results are important for conservation purposes.

Additional comments

My main general concern is that in the Introduction, the authors set up their manuscript by invoking the importance of having behavioral data to validate the use of PAM as a conservation tool (specifically to validate its use as a way to estimate population densities of species that give loud calls). They then provide results of analyses of variation in pant hooting rates in relation to differences in activity state and to variation in vegetation types, etc., and they examine the question of sex differences. They don’t tie this in with the over-arching question regarding the value of PAM. Most of the results are not surprising. The extensive literature on chimpanzee loud calls (pant hoots, specifically) is replete with statements that chimpanzees call most often when they are traveling, when they arrive at food sources, especially fruiting trees, and during feeding bouts. Kallan might have been the first to publish comparative numerical data on male and female calling rates, but Clark (1996, Amer. J. Primatol. 39: 159; cf. Clark & Wrangham, 1994) stated that adult males called significantly more often than subadult males, and that females called at rates comparable to those of subadult males. He further stated that “most adult females called to infrequently to be analyzed statistically” and, in Clark & Wrangham (1994), that “[All-] Female parties called rarely” and did not call on arrival at unoccupied feeding trees. Goodall also referred to sex differences in pant hooting, as did Marler in his early work on vocalizations, albeit without providing the kind of quantitative data the authors have provided in Table 1. For that matter, the data presented here do not confirm that male chimpanzees “vocalize” more than twice as often as females, contrary to the statement on 222-223. That may be correct, but the authors have explicitly not included data on low volume, short-distance calls, so we don’t know whether a sex difference exists for those calls and, if so, what its direction is. Also, while we could expect to find a sex difference in pant-hooting rates (as noted above), the authors have pooled data on these with data on screams and barks, so we don’t actually know the extent of sex difference in each of those three kinds of vocalizations. On 243, the authors note that the bimodal temporal pattern of calling (pant-hooting?) was “similar to what has been reported previously at Issa”. What is really new, then, besides the examination of calling rates in relation to vegetation type? Even there, the authors can only speculate about why the differences might exist.

Back to my initial point: what are the lessons for those concerned with chimpanzee conservation (or conservation of other mammals) on a landscape/habitat scale, especially when the animals in question are not habituated and are difficult to find and follow? What do we gain from the call rate data when it comes to assessing the status of chimpanzees in fragmented habitat outside of protected areas in Tanzania and planning conservation strategy, or for that matter, what lessons does this work have for those working on the conservation of “savanna” chimpanzees in West Africa (or chimpanzees in general)?

·

Basic reporting

Overall, the paper meets the reporting standards.

I found the paper easy to read and thought it was written well. I spotted a few minor typographical/grammar errors, which I note in the general comments to the authors.

As a statistician rather than an ecologist, I am not particularly qualified to comment on the completeness of the cited literature, but I found it interesting to read nonetheless.

The figures, and tables are well presented, and the raw data have been shared.

The paper sets out the aims of the paper nicely in Lines 102-118. This section does a great job of describing the specific hypotheses tested, along with the expected findings in light of the existing literature. The discussion and conclusion return to these hypotheses, providing answers based on statistical analysis of the data.

Experimental design

The description of the statistical analysis would benefit from more detail; I don't think I could replicate the analysis despite being very familiar with generalised linear mixed-effect models and the software they used. I thought the results could also involve a more rigorous interpretation of the parameter estimates. Please find more detail in the general comments to the authors.

Otherwise, the paper meets the experimental design standards. The research questions were well-described, and their surveys appear well-designed.

Validity of the findings

The raw data have been provided. I have a few suggestions regarding the statistical analysis; see my general comments below.

The conclusions appear to be well-stated and interesting to me, although as a statistician rather than an ecologist I am not really qualified to comment.

Speculation is clearly identified throughout the discussion and conclusions.

Additional comments

I thought this was an interesting, well-written, and well-motivated paper. Estimating animal density with acoustic spatial capture-recapture (SCR) methods becomes relatively straightforward once a call rate estimate has been established, which is a central objective of this work. The authors made convincing arguments about why understanding chimpanzee call rate is important in its own right.

I was glad to see the authors use a generalised linear mixed effects model to analyse the call-rate data, and I agree that this is probably the best approach to take. In particular, they have used an appropriate response distribution (negative binomial, to account for overdispersion relative to the Poisson distribution) and have done a good job of checking the model's assumptions.

My main source of confusion about the model is linked to Lines 173-174, regarding the random effect for each 'follow'. What exactly is a 'follow'? Are two observations part of the same 'follow' if they are associated with the same individual chimpanzee? Or only if they are observations associated with the same individual within the same day? Overall, I think random effects in a two-level hierarchy are required: you might expect two observations taken on the same individual on the same day to be more similar than observations taken on the same individual but on different days, because an individual might be more vocal on some days than others. At the same time, you might expect two observations taken on the same individual to be more similar than two observations taken on different individuals, because some individuals might generally be more vocal than others. I suspect having random effects for both days ('follows'?) and individuals will probably do a better job at modelling the dependencies between observations. Moreover, putting some sort of temporal structure on observations taken within the same day would probably be advisable, because two observations taken at similar times are more likely to be similar than observations taken at different ends of the day, over and above the fixed effects due to time of day. I believe adding both levels of random effects, along with the temporal covariance structure, can be achieved easily enough with R package the authors have used, so hopefully this doesn't require too much more work. I think this approach might provide some answers to questions posed in the "limitations" section (Lines 313-315), because it allows explicit estimation of both between-hour variance and between-individual variance.

Here are a few more minor specific comments:

Line 67: Because "calls" is a countable noun, I think "fewer" is better than "less" in this sentence.

Line 79: Although I don't think it was made clear in the Marques et al (2012) paper, they actually estimated call density (calls per unit time per unit space) rather than animal density (animals per unit space).

Lines 96-97: I think "... at different rates depending on sex" sounds a little cleaner than "in function of sex".

Lines 102-118: I thought this section did a great job of describing and summarising this study.

Lines 131-134, and elsewhere: I haven't grasped exactly what the plus/minus symbols mean throughout this paper. In some cases I get the feeling they are proving a total range of the data; for example here I think "mean monthly rainfall of 118.4 +/- 92mm" means that the driest months have 26.4 mm of rain, whereas the wettest have 210.4 of rain. In other cases I got the feeling they were indicating the margin of error of an estimate to provide a (presumably 95%) confidence interval (e.g., Lines 225-226). Providing a range to cover all the data is a very different thing to a range of plausible values for a parameter. I'd recommend avoiding using the plus/minus symbol, and describing in words with more clarity, for example by using sentences like "monthly rainfall in the wet season ranges from 26.4 to 210.4 mm of rain", and "a 95% confidence interval for the mean call rate of adult females is...". Also, the lower limit of the range given for rainfall in the dry season (Line 132; "0.6 +/- 0.9mm) goes negative, which may introduce confusion.

Lines 151-152: There's a missing right parenthesis somewhere in here.

Line 157: The term "scans" first appears here and it isn't ever defined. I'm sure what it means, but it seems to be important. What is a "scan"?

Line 180: Linear models assume the errors are normally distributed, not the residuals. Additionally, GLMs and GLMMs don't even have an assumption that the errors are normally distributed anyway, and so even when the model is correct the residuals may not appear to come from a normal distribution. Although I've never used the package, I think DHARMa is plotting randomised quantile residuals, which are a clever transformation of the residuals that should appear to have been generated by a normal distribution. I recommend rewording this sentence accordingly.

Line 190-191: The fact that call rate appears to vary between individuals verifies that a individual-level random effect should be incorporated in addition to the 'follow' random effect (assuming I am correctly interpreting what a 'follow' is).

Line 201-202: If I understand the model correctly, then the difference in vocalisation rates between males and females is a parameter explicitly estimated by the model, but this statement saying that males call about "twice as often" is quite vague. How about presenting the actual estimated relative difference, along with a corresponding confidence interval? I think this can be achieved as follows: Table 3 suggests that females call exp(-0.817) = 0.44 times as often as males (which indeed roughly matches the "males call twice as often" wording above). A 95% CI for this would be exp(-0.817 +/- 1.96*0.216), or (0.29, 0.67). In other words, with 95% confidence, males call between 1.5 and 3.5 times as frequently as females. Communicating uncertainty around estimates like this in a statistically rigorous way would really strengthen this section.

I also felt that the other parameters estimated by the model could have been interpreted with similar rigour; this section states which variables were significantly related to call rate, but it avoids reporting the size of the effects along with corresponding confidence intervals, which I think would greatly strengthen the results. Fitting a nice model but failing to interpret the estimated effects feels like the story is being left a little unfinished.

Lines 209-213: I wasn't sure why a Kruskall-Wallis test was being used to determine the significance of time-of-day. Couldn't the GLMM model be used to achieve this? Something like likelihood-ratio tests could be used to compare the three time levels to one another, with a manual correction used to account for multiple comparisons. A possible weakness of the Kruskall-Wallis approach is that it ignores dependence between observations made in the same follow, which may invalidate its assumptions. The GLMM deals with this dependence, and therefore seems like a more robust model from which conclusions should be drawn. Perhaps I'm misunderstanding things, in which case I think more detail is required to justify the use of a Kruskall-Wallis test around Lines 184-186.

Line 276-277: I don't think "call rate did not change with feeding behaviour" is quite the correct interpretation; a large p-value fails to reject the null hypothesis, but it does not imply the null hypothesis is true.

Line 297-289: Same as above; a non-significant effect of a swollen female in the party does not imply that there is no difference, only that there isn't evidence a difference exist.

Line 317: I think saying that the data might have "been biased" is a bit negative, and may lead to readers unfairly questioning the validity of this study. As far as I can tell, the data set is great! I think the point being made is that bias can be introduced by a researcher overgeneralising the results of this study to other points in time; this isn't the fault of the data at all, but rather inappropriate conclusions being drawn based on perfectly valid data. I'd recommend rephrasing this to say that conclusions shouldn't be generalised to other scenarios other than that which the data were collected.

Lines 360-361: The common terms are either "spatial capture-recapture (SCR)" or "spatially explicit capture-recapture (SECR)", rather than "spatially capture recapture".

Line 373: Typo; should be "... for their help in the field" rather than "... for their help a in the field".

---

## Round 0.2 · Minor Revisions

Reviewer #1 did not choose to review your paper a second time; in my book, silence (of this duration) is assent. Reviewer #2 wants a few minor changes and is worried about the K-W test---see his detailed comment. This reviewer is obviously enthusiastic about your paper and is well-placed to make suggestions about data analysis.

Please do perform the model-fitting procedure he suggests, and then re-submit the paper. I suspect it'll increase the citations of your paper, in the end. If you object to this further analysis, explain to me the reasons why.

By the way, the basic letter is computer-generated and I cannot modify it. Thus the incongruities between this comment and the letter.

·

Basic reporting

As per my original review, the paper meets the basic reporting standards.

Experimental design

The authors have greatly improved the description of their statistical model. I still have some questions/suggestions about their method, and I think an improvement could be made to strengthen their analysis. I provide further detail in the general comments below.

Validity of the findings

The paper largely meets the validity of findings criteria. I have a minor concern with the conclusion based on the Kruskall-Wallis test, which I outline in the general comments below.

Additional comments

The authors have done a great job with this revision, and the manuscripts has greatly improved. I have a few minor suggestions, listed at the end of this section.

My only substantial comment is about the use of the Kruskall-Wallis test. Thanks to the clear and concise description provided by the authors in their responses to my comments, I now have a much better understanding, which I'll summarise here to ensure I'm on the right page. Their initial attempt to analyse the data involved creating a categorical variable, with a level for every hour of the day, so for example if follows were conducted between 6am-6pm then there would be twelve categories (6-7am, ..., 5-6pm). Unsurprisingly, the large number of parameters required for this approach led to model identifiability issues and convergence failures during model fitting. For their GLMM, they therefore reduced the variable to only three categories. However, they still wished to compare data hour-by-hour, so they used the Kruskall-Wallis test separately.

My concern with this approach is that a Kruskall-Wallis test still assumes statistical independence, despite the authors acknowledging that data from two different follows are not statistically independent; their GLMM was constructed specifically to account for this dependence. If the independence assumption is violated, then it's possible for p-values to detect spurious effects at a rate higher than the selected significance level (i.e., a true null hypothesis will be rejected greater than 5% of the time), resulting in misleading conclusions.

Additionally, splitting a continuous explanatory variable (such as time) into a number of categories, be it three or twelve, is not generally consiered good practice; see textbooks such as Generalized Additive Models by Simon Wood and Vector Generalized Linear and Additive Models by Thomas Yee. A better approach in this situation is to fit the effect of time as a smooth function, for example using a spline or a generalised additive model. I think it's easy enough for the authors to do exactly this with their data set via the mgcv package using code something like the following:

fit.gam <- gam(y ~ s(individual, bs = "re") +
s(date, bs = "re") +
s(hour, bs = "cc") +
[any other fixed effects, like group size],
knots = list(hour = c(0, 24)),
family = "nb", data = df)

Then using plot(fit.gam) you can visualise the effect of hour on the underlying call rate as a smooth function, allowing you to see the spikes at dawn and dusk (for example). This model will also return a p-value for the significance of the time-of-day variable, just like the Kruskall-Wallis test, but is more appropriate because it also acknowledges dependencies among the observations. Note that using s(x, bs = "re") in mgcv is equivalent to using (1 | x) for random intercepts in lme4. I think this would be both more appropriate and more insightful than (1) the current GLMM with the three-level categorial variable for time and (2) the Kruskall-Wallis test. Generating conclusions from a single model is also conceptually neater than using two separate models.

To demonstrate this idea a little more clearly, I simulated data and fitted the model under a slightly simpler scenario (only individual random effects and no other explanatory variables other than hour of day), and the script is available here: http://bcstevenson.nfshost.com/chimpanzee-gams.r.

Minor comments follow.

L53: It might be best to define call density as the number of calls emitted per unit time *per individual* (just to avoid misinterpretation of number of animals emitted per unit time across the whole population).

L86: I think it's important to be clear about what you mean by a "loud call". How do you know if a call is "loud" or not? It seems problematic to define whether or not a call is "loud" based on the loudness of the call at its source (e.g., measured in dB), because you don't directly observe this variable. After reading on, I think the definition appears later on L160-162: a "loud" call is one that involved at least a scream, a pant-hoot, or a bark. Is that right? If so, I think the sentence here on L86 feels a little out of place, because using only "loud calls" and defining them in this way is particular to this very study, whereas this paragraph here is about PAM in general without focus on any particular species.

L106: Marques et al (2013) might be a better reference here because their study was about passive acoustic monitoring in general, whereas Stevenson et al (2015) is focused only on SCR.

L153-154: This is a nice definition and clarifies my confusion in my last review.

L180-182: I suggest "dependency among follows on the same day", because it's possible to misinterpret "at the same time" to mean the very same instant in time, rather than simply two follows in different hours of the same day.

L298, 305: In these lines the authors assert that they found that some explanatory variables "do not impact call rate", but, similarly to my previous review, these statements still sound like misinterpretations to me. I think they are being made due to large p-values, or possibly because explanatory variables were not included in the most parsimonious model. However, neither of these findings imply that the variable has no effect (or impact), only that there is no evidence that it does have impact. If the effect of a variable is very small it still has an "impact", but without sufficient data a model will not be able to detect its effect. Absence of evidence of an effect is not evidence that the effect is absent.

---

## Round 0.3 · accepted · Accept

Congratulations. Good paper!